# *Pseudogymnoascus destructans* Transcriptional Response to Chronic Copper Stress

**DOI:** 10.3390/jof11050372

**Published:** 2025-05-13

**Authors:** Saika Anne, Maranda R. McDonald, Yuan Lu, Ryan L. Peterson

**Affiliations:** 1Department of Biology, Texas State University, San Marcos, TX 78666, USA; 2Department of Chemistry and Biochemistry, Texas State University, San Marcos, TX 78666, USA; 3Institute for Molecular Life Sciences, Texas State University, San Marcos, TX 78666, USA; y_l54@txstate.edu

**Keywords:** white nose syndrome, copper, transcriptomics, metal-ion homeostasis, ionophore, metallophore

## Abstract

Copper (Cu) is an essential metal micronutrient, and a fungal pathogen’s ability to thrive in diverse niches across a broad range of bioavailable copper levels is vital for host colonization and fungal propagation. Recent transcriptomic studies have implied that trace metal acquisition is important for the propagation of the white nose syndrome (WNS) causing fungus, *Pseudogymnoascus destructans*, on bat hosts. This report characterizes the *P. destructans* transcriptional response to Cu-withholding and Cu-overload stress. We identify 583 differently expressed genes (DEGs) that respond to Cu-withholding stress and 667 DEGs that respond to Cu-overload stress. We find that the *P. destructans* Cu-transporter genes CTR1a and CTR1b, as well as two homologs to *Cryptococcus neoformans* Cbi1/BIM1 *VC83_03095* (BLP2) and *VC83_07867* (BLP3), are highly regulated by Cu-withholding stress. We identify a cluster of genes, *VC83_01834* – *VC83_01838*, that are regulated by copper bioavailability, which we identify as the Cu-Responsive gene Cluster (CRC). We find that chronic exposure to elevated copper levels leads to an increase in genes associated with DNA repair and DNA replication fidelity. A comparison of our transcriptomic datasets with *P. destructans* at WNS fungal infection sites reveals several putative fungal virulence factors that respond to environmental copper stress.

## 1. Introduction

The redox properties of copper (Cu) make it an important metal micronutrient to support eukaryotic life [1,2]. Copper metalloenzymes participate in a wide range of biological processes [3], ranging from electron transfer [4], antioxidant defense [5,6], and small-molecule activation [7,8] to iron acquisition [9]. While trace copper levels are essential for maintaining cellular function, the over-accumulation of copper can be toxic, leading to mitochondrial dysfunction and deleterious metal-mediated oxidative stress [10,11]. In fungi, the copper-sensing transcription factor pair Mac1/Ace1 or single transcription factor Cuf1 help regulate essential Cu-acquisition or Cu-detoxification genes to maintain Cu homeostasis [12,13,14]. Within the context of microbial infectious disease, the ability of fungal pathogens to scavenge essential metal ions such as copper from their infected host is critical for host colonization and pathogen propagation [15,16]. Animal hosts can sequester labile metal pools to starve invading microorganisms through a process known as nutritional immunity utilizing high-affinity metal-binding proteins such as calprotectin or the family of S100 proteins [17,18,19]. The toxic nature of Cu is also exploited by animal hosts in certain niches to kill invading microbial pathogens. Specifically, ATP7a Cu-transporters within maturing macrophage phagolysosomes can bombard engulfed pathogens with excess toxic Cu-ions to boost macrophage-killing efficiency [20]. Thus, the ability of fungal pathogens to regulate Cu-importing and Cu-exporting machinery is important if they are to thrive on animal hosts.

Baker’s yeast *Saccharomyces cerevisiae* provided early insights into the basic Cu-cell biology of fungi and higher eukaryotes [21,22]. However, recent advancements in the Cu-cell biology of human-infecting fungal pathogens such as *Candida* [23,24], *Aspergillus* [25], and *Cryptococcus* [26,27], have revealed diverse strategies beyond that found in *S. cerevisiae* and other higher eukaryotes for managing intracellular and extracellular copper levels. In *Candida albicans*, the transcription factor Mac1 inhibits Cu/Zn-SOD transcription to help divert cellular copper pools for respiration [24]. *Aspergillus fumigatus* can secrete small-molecule isocyanide calkophores to scavenge extracellular copper pools, which may also play a role in Cu metabolism in a wide range of other fungal species [25,28]. Meanwhile, *Cryptococcus neoformans* has been shown to use the Cu-scavenging protein CBI1/BIM1 to mobilize extracellular copper and deliver it to high-affinity Cu-transporter (Ctr) proteins [26]. However, much less is known about the basic metal cell biology of other animal infecting fungal pathogens that are less prone to infecting human hosts.

*Pseudogymnoascus destructans* is the fungal pathogen that is responsible for white nose syndrome (WNS) in bats [29,30]. Fungal infection causes severe burn-like skin lesions primarily located on a hibernating bats’ nose, muzzle, and wings when their immune system is less active and they are in an immunosuppressed state [31]. *Pseudogymnoascus destructans* has a unique evolutionary history thought to have been a pathogenic soil-dwelling plant-associated fungi that later evolved to thrive on vulnerable bat hosts [32]. The ability of *P. destructans* to thrive in diverse soil, plant, and animal niches provides a unique opportunity to understand how fungi can adapt to secure essential metal micronutrients from diverse hosts or environmental niches.

Previous transcriptomics studies of active fungal infection sites on WNS bats display hallmarks of fungal-host adaptations that are like those found in other mammalian hosts [33,34,35]. Field and coworkers identified elevated transcripts for bat-host S100 proteins as well as a collection of *P. destructans* high-affinity Fe-, Cu-, and Zn-metal transporters [34]. Together, these transcriptional studies suggest that the metal bioavailability of the host–pathogen interface is devoid of essential metal ion micronutrients for the invading *P. destructans* fungi. While the genes involved in the *P. destructans* Cu-stress responses can be inferred through these transcriptomic studies at the WNS infection sites, other contributing stress factors at the host–pathogen interface may impact global *P. destructans* transcriptional levels. How *P. destructans* adapts to metal micronutrient sequestration and overload stress is not known.

This report systematically investigates the *P. destructans* Cu-stress transcriptional response under laboratory conditions using chemically defined media to simulate Cu-withholding and Cu-overload stress. Global transcriptional analysis reveals unique transcriptional responses for *P. destructans* for Cu-withholding stress and Cu-overload stress that are unique to control growth conditions. Our data are consistent with *P. destructans* utilizing two high-affinity CTR and two Bim1-Like Proteins (BLPs) to secure essential copper from its environment. Additionally, we identify a curious cluster of genes that we have named the Cu-Responsive gene Cluster (CRC) that may encode for genes involved for the biosynthesis and trafficking of a small-molecule ionophore under Cu-restrictive growth conditions. Under Cu-overload stress, we find enrichment of *P. destructans* DNA replication and DNA repair pathways, suggesting that Cu-overload stress may lead to *P. destructans* genome instability. Finally, our data show that copper stress can induce many putative *P. destructans* virulence factors. Together, this study represents the first systematic investigation into the transcriptional response of *P. destructans* to Cu stress and may provide insight into *P. destructans* biology relevant to WNS disease.

## 2. Materials and Methods

### 2.1. Media Preparation, Propagation, and Imaging of Pseudogymnoascus destructans

The *Pseudogymnoascus destructans* strain (MYA4855) was purchased from the American tissue culture collection (ATCC strain number 20631-21) and used for all experiments. The long-term propagation of *P. destructans* was carried out using solid yeast extract–peptone and glucose (YPD) plates at 15 °C and consisted of 10 g/L of yeast extract, 20 g/L of peptone, 20 g/L glucose, and 15 g/L of agar. Metal growth studies were performed using chemically defined growth plates (SC-Ura) which were made using Milli-Q water and 1.8 g/L of yeast nitrogen base, 5 g/L ammonium sulfate (NH_4_SO_4_), 20 g/L glucose, 100 mg/L histidine, 1.7 g/L of the SC-Ura-His amino acid supplement mixture (Sunrise Scientific, Knoxville, TN, USA), and 15 g/L agar. Before autoclaving one pellet (~200 mg) of NaOH was added to the SC-Ura media prior to autoclave sterilization. The addition of sterile Cu-sulfate and Bathocuproine sulfonic acid sodium salt (BCS; Acros Organics, Geel, Belgium) was added to the SC-Ura growth media using a 125 mM stock solution after autoclave sterilization at a temperature of approximately 50 °C to a final BCS concentration of 800 µM or 500 µM unless explicitly noted otherwise.

The inoculation of SC-Ura experimental growth plates was performed as follows: *Pseudogymnoascus destructans* conidia and mycelium were harvested from a working YPD plate, grown for two weeks at 15 °C by adding 2 mL of 1× TE buffer to the plate and gently rubbing the mycelium mass with an inoculation loop. The resulting cell suspension was transferred and passed through a sterilized pasture pipette containing a plug of Miracloth (EMD Millipore Corp, Burlington, MA, USA) to isolate *P. destructans* conidia. One hundred microliters of the resulting filtrate was used to inoculate experimental SC-Ura plates.

Imaging of harvested *P. destructans* was conducted as described above. Fungal spores were mounted with ProLong gold mounting media (Invitrogen, Waltham, MA, USA) and imaged on an inverted Zeiss Axio observer microscope with an Axiocam 820 camera (ZEISS, Oberkochen, Germany) under brightfield illumination.

### 2.2. Pseudogymnoascus destructans RNA Extraction and Isolation

Transcriptomic studies were performed on *P. destructans* fungi grown for 10 days at 15 °C on SC-Ura plates alone or supplemented with either the copper chelator BCS (800 µM) or CuSO_4_ (500 µM). Fungal cells, including conidia and mycelium, were suspended in 2 mL of TE buffer by gently rubbing the mycelium mass with a sterile inoculating loop. The resulting cell suspension was transferred to a 2 mL microcentrifuge tube and pelleted for 5 min at 16,000× *g*. The cells were washed twice with 1 mL of TE buffer. The resulting cell pellet was snap-frozen in dry ice and stored at −80 °C for later use. Total RNA was extracted from *P. destructans* cells using TRIZOL reagents using the manufacturer’s recommended procedure (ThermoFisher Scientific, Waltham, MA, USA). Briefly, 1 mL of TRIZOL reagent was added to *P. destructans* cells along with approximately 100 μL of 0.5 μM zirconium beads (ZB-05; Next. Advanced) and homogenized at 4 °C in a bullet blender (Speed 12 for 5 min). The resulting cell lysate was incubated on ice for 5 min. The cell lysates were then thoroughly mixed with 0.2 mL chloroform and incubated for 2 to 3 min before centrifugation at 12,000× *g* for 15 min. The upper aqueous layer was transferred to a new microcentrifuge tube, and 500 μL of isopropanol was added. After mixing, the solution was incubated on ice for 10 min and centrifuged at 12,000× *g* at 4 °C. The supernatant was discarded, and 1 mL of 75% ethanol was added, vortexed, and centrifuged at 7500× *g* for 5 min at 4 °C. The supernatant was discarded, and the white pellet was air-dried for 5–10 min. The RNA (pellet) was resuspended in 50 µL of 1× TE buffer and incubated for 15 min at 55 °C in a hot water bath. Finally, the concentration of the RNA was determined using nanodrop and analyzed using an Agilent bioanalyzer (Santa Clara, CA, USA). Then, RNA was stored at −80 °C. All RNA samples were sent to Innomics, Inc., (Sunnyvale, CA, USA) for the polyA mRNA enrichment, sequencing library construction, indexing, and RNA sequencing.

### 2.3. Differential Gene Expression Under Copper Stress

Sequencing reads were delivered by sequencing facility through a cloud server on Amazon Web Services (AWS). Sequencing reads were processed using ftastx_toolkit (https://www.hannonlab.org/resources/ accessed on 8 November 2023) to trim off any adapter sequences, low-quality base calls, and filter low-quality sequencing reads. Processed sequencing reads were mapped to the *P. destructans* reference genome (Accession number: GCF_001641265.1_ASM164126v1_genomic) using Tophat2 (version 2.0.13) [36]. Gene expression profiling was subsequently conducted using the featureCounts (subRead package v.1.4.6) [37] function of the subread package to quantify sequencing reads mapped to exons guided by a genome annotation file corresponding to the genome.

The edgeR package (version edgeR_4.0.16) available in Bioconductor was used to identify the total number of differentially expressed genes between biological conditions. Where differentially expressed genes (DEGs) were determined by |log_2_FC| > 1, FDR < 0.05 (Appendix A). Subsequential global analysis of DEGs meeting this criterion and a logCPM > 1 were retained for further analysis, ensuring a focus on robustly expressed candidates in the subsequent stages of investigation (Appendix A).

We used Venny 2.1 to construct intersections, logical relationships, and commonalities within treatment groups in comparison to control groups within *P. destructans* samples.

### 2.4. Gene Ontology (GO) and GO Enrichment Analyses

We extracted GO terms associated with biological processes for each DEG from JGI (https://mycocosm.jgi.doe.gov/Pseudest1/Pseudest1.home.html, accessed on 21 March 2024). For each GO term, we counted the number of DEGs that have the GO term, the number of DEGs that have the same particular GO term, the total number of genes that have the GO term, and the number of genes in the genome sharing the same particular GO term, and conducted hypergeometric test to identify over-represented GO terms (*p* < 0.05).

### 2.5. Identification of Pseudogymnoascus destructans Gene Homologs

The basic local alignment search tool (BLAST) at the NIH National Center of Biotechnology Information (https://blast.ncbi.nlm.nih.gov/Blast.cgi, accessed on 6 November 2024) was used to identify homologs for *P. destructans* proteins.

### 2.6. RNA Sequencing Datasets

All RNA sequencing datasets have been deposited into the NIH GEO database (https://www.ncbi.nlm.nih.gov/geo/, accessed on 29 April 2025) with the accession numbers GSM8964574-GSM8964596. 

## 3. Results

### 3.1. Pseudogymnoascus destructans Growth Under Copper Stress

To characterize the ability of *P. destructans* to grow under extremes in Cu bioavailability in SC-Ura growth media, we screened the effect of added Cu sulfate and the copper chelator BCS on *P. destructans* growth. We tested varying concentrations of supplemental copper sulfate (i.e., 100–500 μM) to simulate Cu-overload growth conditions and added the copper chelator BCS (i.e., 0.2–1 mM) to simulate Cu-withholding growth conditions (Appendix A). We observed that *P. destructans* had a high tolerance to Cu, with no growth inhibition observed at concentrations up to 500 μM supplemental Cu-sulfate. The gray-green color of the *P. destructans* hyphal mass when grown under copper-overload conditions may indicate an increase in laccase-produced fungal melanin due to cellular copper homeostasis or a developmental regulator [38], whereas modest *P. destructans* growth inhibition can be observed with BCS at concentrations > 800 μM. Based on these experiments, we selected the 500 µM CuSO_4_ to stimulate Cu-overload conditions and 800 µM copper chelator BCS to stimulate Cu-withholding conditions for transcriptomic analysis (vide infra).

Fungi cultured under control and Cu-withholding stress conditions for 10 days had tan mycelium in contact with the growth plate and were off-white on the top of the growth plate (Figure 1a,c). On the other hand, under high copper conditions, *P. destructans* mycelium turned dark gray-green both on the top and bottom of the growth media plate (Figure 1b). Microscopic examination of *P. destructans* conidia revealed that copper stress growth conditions did not lead to noticeable changes in their characteristic asymmetric curved conidia morphology [29].

### 3.2. Differential Gene Expression in Response to Copper Stress

To gain insight into *P. destructans* copper cellular homeostasis and the pathways modulated by copper withholding and overload stress, we performed global gene expression analysis of mRNA transcripts of *P. destructans* conidiophores grown under control (i.e., SC-Ura), Cu-withholding (i.e., SC-Ura + 800 µM BCS) and Cu-overload (i.e., SC-Ura + 500 µM CuSO_4_) conditions. Using clustering by edgeR, we found similar patterns of *P. destructans* gene expression within each experimental group. Multi-dimensional scaling (MDS) profiles of Cu-withholding versus control and Cu-overload versus control treatments are displayed in Figure 2a,b, respectively. The sample distribution patterns suggest that the differences in gene expression profiles between the two experimental groups are due to variations in *P. destructans* environmental growth conditions.

Analysis of copper-withholding vs. control transcription profiles revealed a total of 583 genes that were significantly differentially expressed (*p* < 0.05; |log_2_FC| > 1) between experimental growth conditions (Figure 2c). Within this dataset, 399 genes were upregulated, and 184 genes were downregulated vs. the control samples. In contrast, analysis of copper-overload vs. control conditions revealed 667 differentially expressed genes (*p* < 0.05, |log_2_FC| > 1) that were significant between the two experimental groups (Figure 2d). A total of 542 genes were upregulated, and 125 genes were downregulated between copper-overload and control culture conditions. Further global analysis was performed on differentially expressed genes with *p* < 0.05, |log_2_FC| > 1, and total normalized read counts of >10 counts per million (CPM) under at least two growth conditions.

A Venn diagram was constructed to analyze common and unique differentially expressed genes (DEGs) upon Cu-withholding vs. Cu-overload growth conditions (Figure 3a). We observe that 334 genes are uniquely upregulated under copper overload (UR C vs. Cu), while 175 upregulated DEGs are also upregulated under copper-withholding growth conditions (UR C vs. BCS). A unique cluster of seven DEGs is upregulated under Cu-overload and downregulated under Cu-withholding conditions (Figure 3a and Appendix A). We anticipate these seven genes may protect for Cu stress or be involved in the *P. destructans* Cu-detoxification pathways. These seven genes encode for a major facilitator family (MFS) transporter (*VC83_08519*), a plasma membrane-associated MARVEL Domain protein (*VC83_03991*), three predicted secreted proteins (*VC83_05770*; *VC83_03107*; *VC83_06529*), and two additional proteins with no clear protein homolog (*VC83_07925*; *VC83_07926*) (Appendix A).

The cluster of nine shared DEGs that are downregulated upon Cu overload and upregulated under Cu-withholding stress expression bear hallmarks of known genes involved in trace metal ion-acquisition pathways (Figure 3b) [39,40,41]. This behavior suggests that Cu-bioavailability regulates gene transcript levels, and these genes play critical roles in the *P. destructans* Cu-stress response mechanism. Six of the nine genes in this cluster include homologs to the family of high-affinity copper transporters (CTRs) [42,43,44], FRE metalloreductase [45,46], and the extracellular Cu-scavenging protein *C. neoformans* CBI1/BIM1 [26,47]. The three genes *VC83_01835*, *VC83_01836*, and *VC83_01837* encode for a Major Facilitator Superfamily (MSF) transporter, opine dehydrogenase, and *S*-adenomethionine-utilizing enzyme, respectively. The latter three genes are associated with the production and trafficking of opine metallophores commonly found in bacteria [40,48,49,50]. This cluster of nine genes is consistent with the notion that under low Cu-stress, *P. destructans* can utilize high-affinity Cu-trafficking proteins and secreted small molecules to acquire trace copper from the environment to fulfill its nutritional copper requirements.

### 3.3. Comparative Gene Ontology Analysis of P. destructans Under Cu-Withholding and Cu-Overload Conditions

To gain insight into the global function of *P. destructans* DEGs regulated by Cu-withholding and Cu-overload stress, gene ontology enrichment analysis vs. control growth conditions was performed and are displayed in Appendix A, respectively. Copper-withholding stress leads to 24 biological pathways that are significantly enriched with *p* < 0.05 (Table 1). In contrast, high copper stress leads to 35 significant enriched biological pathways with *p* < 0.05 (Table 2). The two growth conditions share 11 enriched pathways. These shared pathways concentrate on the melanin metabolic processes, electron transport, carbohydrate transport, copper transport, and amino acid transport. While some of the gene ontology between the two copper stress conditions is shared, most are unique and are consistent with the notion that *P. destructans* adapts uniquely to chronic Cu-withholding and Cu-overload stress.

To further understand the *P. destructans* copper stress response, we performed a targeted comparison analysis of *P. destructans* DEGs against model fungus *Saccharomyces cerevisiae* and the three pathogenic fungi *Candida albicans*, *Aspergillus niger*, and *Cryptococcus neoformans*. Our comparison focused on genes associated with the metabolic processes including melanin metabolic processes (GO:0006582), superoxide metabolic processes (GO:0006801), response to oxidative stress (GO:0006979), carbohydrate transport (GO:0008643), copper ion transport (GO:0006825), and electron transport (GO:006118). Significant regulated genes associated with Cu-withholding stress and their predicted subcellular localization are shown in Appendix A, while significant regulated genes associated with Cu-overload stress are displayed in Appendix A. Our results indicate that *P. destructans* adapts similarly to other fungal species with more characterized copper stress responses.

### 3.4. Putative Virulence Factors Controlled by Copper Stress

Previous transcriptomic studies at active sites of *P. destructans* infections sites on WNS-positive bats by Field and coworkers identified candidate fungal virulence factors [34]. Cross-referencing their list of candidate fungal virulence factors, we find that several genes also respond to copper stress. We identify 15 genes that exhibited significant changes in transcript levels under low copper stress and 13 genes that exhibited significant changes in their transcripts under high-copper-stress conditions (Table 3). Specifically, eight genes (*VC83_02553*, *VC83_01046*, *VC83_08187*, *VC83_06435*, *VC83_04094*, *VC83_06039*, *VC83_09074*, and *VC83_02181*) exhibited elevated transcripts under both low and high Cu conditions. Conversely, seven genes (*VC83_00970*, *VC83_01360*, *VC83_00191*, *VC83_07867*, *VC83_00261*, *VC83_05292*, and *VC83_01624*) showed exclusive upregulation under Cu-withholding conditions. Additionally, two genes, *VC83_09076* and *VC83_06062*, responded uniquely to high-copper-stress conditions. Notable examples of these genes include those involved in the heat shock response, such as *VC83_02553* (Alpha-crystallin domain, HSP) and *VC83_00970* (Heat shock protein 78, mitochondrial), as well as transporters like *VC83_01360* (ZIP Zinc transporter), *VC83_00191* (Ctr copper transporter family), *VC83_04094* (ATX1; HMA; heavy-metal-associated domain) and *VC83_07867* (a Bim1-like protein (BLP)). These findings suggest that *P. destructans* DEGs that respond to copper stress under laboratory conditions are relevant to the white nose syndrome (WNS) state on its native bat-host.

## 4. Discussion

Copper is an essential metal micronutrient needed to support eukaryotic life, yet it plays a dual role in fungal pathobiology, acting both as a scarce resource as well as a toxic agent at high concentrations [15,16]. In fungi, two classes of Cu-responsive transcription factors regulate the machinery needed for trace Cu acquisition and Cu detoxification to ensure efficient cellular propagation: the reciprocal pair of Mac1/Ace1 [12,14] transcription factors and single component Cuf1 [51,52] transcription factor. For fungal pathogens, extremes in Cu-bioavailability can be encountered over the course of the mammalian host infection [15], and thus, fungal pathogens must adapt to manage this dynamic metal resource within their host niche. *Pseudogymnoascus destructans* is a curious fungal pathogen with a unique evolutionary trajectory and is currently thought to be an ancient soil-dwelling plant pathogen that evolved to thrive on susceptible bat hosts. Recent transcriptomic studies by Field and coworkers suggest that trace micronutrient acquisition pathways are associated with bat-host colonization and identified several fungal genes involved in zinc, iron, and copper homeostasis as possible fungal virulence factors [34]. This study aims to describe the *P. destructans* transcriptional responses to low- and high-copper-stress growth conditions to gain insight into how *P. destructans* may adapt to changing copper levels at the WNS host–pathogen interface or cave-hibernacula soil niches.

### 4.1. Copper Stress Metabolic Adaptations

While gross changes in fungal spore morphology have been described in *C. neoformans* upon BCS-mediated Cu-withholding stress [53], we do not observe these changes in *P. destructans* spores, which have maintained their size and distinct curved conidia shape. However, transcriptional analysis of the most abundant 50 DEGs (Figure 4a,b) reveals significant modifications within the *P. destructans* cellular transcriptome, which are unique to copper-withholding and copper-overload stress.

### 4.2. P. destructans Adaptation Under Cu-Withholding Stress Conditions

The family of high-affinity Cu-transporters (CTRs) facilitates the movement of reduced Cu-ions across biological membranes under Cu-limiting growth conditions. While Ctr1 proteins can act independently to selectively import Cu ions from the extracellular environment, more recently, in *C. neoformans*, the extracellular scavenging protein, CBi1/BIM1, has been shown to assist in extracellular copper capture, both in in vitro and in vivo models, and has also been shown to participate in fungal cell wall integrity [26,47,54]. The *P. destructans* genome encodes for two Cu transporters, *VC83_00191* (CTR1a) and *VC83_04814* (CTR1b), as well as three BIM1-like proteins (BLPs) *VC83_02818* (BLP1), *VC83_03095* (BLP2), and *VC83_07867* (BLP3) [55]. Copper-withholding stress leads to a significant 12.0- and 18.6-fold increase in CTR1a and CTR1b respective transcripts versus control growth conditions. The CTR1a isoform transcripts are approximately four times more abundant in the CTR1b isoform, suggesting that both isoforms may be involved in trace Cu-acquisition. These observations are consistent with previous work by our group, which identified *P. destructans* CTR1a as a functional high-affinity Cu-transporter that is transcriptionally activated and expressed under Cu-limited growth conditions [55]. We also observe a significant 3.8 and 26.3-fold increase in transcript levels for two of three BLP isoforms, *VC83_03095* (BLP2) and *VC83_07867* (BLP3), respectively, under copper-withholding growth conditions. However, minimal alterations in the BLP1 isoform (i.e., *VC83_02818*) transcript levels are observed under the varied Cu-stress growth conditions tested in our study. These observations are consistent with the notion that *P. destructans* may employ multiple BLP-isoforms to scavenge Cu-ions from is environment.

The three predicted Fre-metalloreductases enzymes (MRed) *VC83_03096*, *VC83_08787*, and *VC83_01834* are also significantly upregulated upon Cu-withholding stress, which may facilitate the Cu^2+^ to Cu^+^ reduction chemistry at the cell surface for the efficient transport of Cu^+^ ions through CTR channels. In Baker’s yeast, *Saccharomyces cerevisiae*, FRE1 and FRE2 are the primary genes that facilitate reductive Fe- and Cu-import [56,57]. FRE homologs in other pathogenic fungi such as Fre7 in *Candida albicans* [58] and Fre4/Fre7 in *Cryptococcus neoformans* [59] can respond to Cu-withholding stress. An overview of the putative *P. destructans* CTR/BLP/MRed enzymes participating in high-affinity Cu-import is overviewed in Figure 5. Together, our data are consistent with the notion that the *P. destructans* copper-withholding stress response involves the transcriptional regulation of both CTR1 isoforms as well as two CBi1/BIM1-homologs to scavenge trace copper ions from their extracellular environment.

Identification of a putative novel fungal high-affinity metal acquisition pathway in *P. destructans*

Pathogenic microbes can secrete metal-binding small molecules to capture essential metal micronutrients. The most well studied are the Fe-binding siderophores [60], although Cu-binding calkophores [61] and Zn-binding zincophores [62,63,64,65] have been reported. We identify a cluster of five genes *VC83_01834*–*VC83_01838*, which show differential gene regulation displaying reciprocal transcriptional regulation in response to copper stress growth conditions (Figure 3 and Appendix A). We identify *VC83_01834* –*VC83_01838* as the Copper-Responsive gene Cluster (CRC) (Figure 5). The four genes *VC83_01834*, *VC83_01835*, *VC83_01836*, and *VC83_01837* segregate in a cluster of DEGs, which are significantly upregulated under Cu-withholding stress and repressed under Cu-overload conditions (Figure 3a,b). We have included the gene *VC83_01838*, which encodes for a cytochrome P450 monooxygenase enzyme, in the CRC gene cluster because it remains a significant DEG under both Cu-withholding and Cu-overload stress.

The genes encoded in the CRC may participate in the production and trafficking of a calkophore or metallophore small molecule. To our knowledge, *P. destructans* cannot produce isocyanide Cu-binding calkophores that other fungi can secrete due to the lack of ISP encoded in its genome [28]. However, the *P. destructans* genes *VC83_01835*, *VC83_01836*, and *VC83_01837* encode for homologous proteins found in the bacterial operons responsible for the production and trafficking of the opine-metallophores staphylopine [50] and pseudopaline [49], respectively. The presence of the metallo-reductase (*VC83_01834*) and cytochrome P450 monooxygenase genes (*VC83_01838*) in this gene cluster is less clear, but may be required for cell surface reduction chemistry needed for metal uptake or the maturation of the small-molecule metallophore. Minor modifications of the opine-metallophore core structure via diversification of amino acids and metabolites are known to generate species-specific metallophore small-molecules [66]. Secreted *P. destructans*-derived small molecules, including riboflavin [67] and triacetlyfarcine C [68], have been observed at the host–pathogen interface on WNS-positive bat samples. It is possible that *P. destructans* may have evolved a novel Cu-trafficking pathway involving a small molecule as an alternative or redundant pathway to the resource-intensive protein mediate BLP/CTR pathway (vide supra) [26]. Future studies that focus on the transcriptional regulation of the CRC to other trace metals (i.e., Fe and Zn), as well as validating CRC protein levels under Cu-withholding stress, will be needed to help elucidate its role in *P. destructans* metal-cell biology and its possible role in virulence.

Remodeling of *P. destructans* superoxide dismutase (SOD) enzymes under Cu-withholding stress

The intracellular bimetallic Cu/Zn SOD can act as a cytosolic copper buffer to facilitate extremes in Cu-homeostasis while protecting against cytoplasm superoxide damage [69]. In the absence of Cu, some animals infecting pathogenic fungi can employ dedicated cytosolic targeting Mn-SODs (i.e., the Mn-SOD3 in *Candida albicans*) or generate alternative cytosolic targeting Mn-SOD2 transcripts to maintain cytosolic antioxidant activity when cellular Cu-levels are sparse [23,27]. The *P. destructans* genome encodes for two Cu-containing SODs and two Mn-SOD enzymes. The transcriptional regulation of the four SOD enzymes under Cu-stress growth conditions is shown in Appendix A. The two Cu-containing SODs variants include the bimetallic cytosolic Cu/Zn SOD (*VC83_07077*) and a secreted GPI-anchored Cu-only SOD (*VC83_08495*), whereas the two Mn-SOD isoforms found in the *P. destructans* genome encode for a mitochondrial-targeting Mn-SOD2 (*VC83_07362*) variant and cytoplasmic Mn-SOD3 (*VC83_02616*) (see Appendix A for homology and Appendix A for SOD3 localization assignment). In our study, we observe a 2.1-fold reduction in Cu/Zn SOD (*VC83_07077*) transcripts and a 2.3-fold increase in Mn-SOD3 (*VC83_02616*) transcripts. We anticipate that this reciprocal SOD transcriptional regulation is needed to divert cellular copper pools to another Cu-enzyme, such as Cytochrome *c* oxidase or FET3 [70], to maintain efficient cellular respiration or Fe-acquisition. However, no significant change in transcript number or isoform for the mitochondrial localizing Mn-SOD2 *(VC83_07362*) transcript levels is observed under the conditions tested. The redistribution of intracellular SOD transcripts upon Cu-restrictive growth conditions may be advantageous, allowing *P. destructans* to maintain cellular redox homeostasis when propagating in Cu-poor soils or allowing it to overcome the metal restrictive host defense mechanisms.

To our surprise, both copper-withholding and copper-overload stress conditions led to significant increases in transcript levels for the extracellular Cu-only SOD (*VC83_08495*) when compared to control growth conditions, suggesting that this extracellular enzyme is important for adapting to both copper-withholding and copper-overload stress. Cu-only SOD isoform transcripts in other filamentous fungi, such as *C. albicans,* have been shown to have distinct expression profiles that are associated with their host niche, such as biofilms for SOD6 [71], macrophages for SOD5 [72], or Fe-poor growth conditions for SOD4 [73]. Together, the transcriptional activation patterns of the four superoxide dismutase enzymes in *P. destructans* appear to mimic that of the pathogenic fungi *C. albicans,* which adapts its SOD transcriptional levels to cope with bioavailable copper levels and stress [23]. Future studies will be needed to determine if these transcriptional observations are also representative at the SOD protein and activity levels.

Notable proteins involved in metal ion homeostasis and Cu-withholding stress adaptations

A putative zinc transporter *VC83_01360*, a *ZRT1* homolog, is also upregulated under Cu-withholding stress. Previous transitional studies have associated *VC83_01360* with fungal virulence on infected bat-hosts [34]. In vitro experiments on *Candida albicans* with the biologically relevant metal scavenger calprotectin in serum-based growth media have shown a decrease in cellular copper levels and have been shown to induce *ZRT1* and *PRA1* transcripts while preserving cellular Zn levels [15]. This may suggest that Cu-withholding stress may induce Zn-acquisition machinery in *P. destructans*. However, future systematic studies in *P. destructans* are still needed to fully elucidate the regulatory crosstalk between Cu and other trace metal ion micronutrients.

Several other biological processes are impacted by Cu-withholding stress, with some genes implemented in *P. destructans* virulence on bat-hosts. Biological processes associated with electron transport are highly representative, with 15 genes identified as being altered by Cu-withholding stress (Table 1; Appendix A). Alterations in electron transfer processes may be anticipated due to copper’s role in electron transport protein active sites and the reductive mechanisms for high-affinity Cu-import. The dihydroxynaphthalene (DHN) melanin pathway is one unexpected pathway that was impacted upon both Cu-withholding and Cu-overload stress. Specifically, secreted scytalone dehydratase gene (*VC83_07909*) transcripts are elevated 35-fold and 6-fold under Cu-withholding and Cu-overload conditions, respectively. The production of melanin under Cu-stress may be a defensive mechanism to protect against the environmental stress of added Cu or nutrient starvation. Melanin produced by the DHN pathway has been shown to protect *Aspergillus fumigatus* against macrophage and neutrophil killing by interfering with phagolysosome maturation [74]. A similar protective role for DHN was recently described for *P. destructans* with an immortalized bat keratinocyte model system [75]. We anticipate that the gene *VC83_07909* may be involved in DHN melanin production and participate in both copper nutrient stress resistance and bat-host colonization.

### 4.3. P. destructans Response to Copper Overload Conditions

In *Saccharomyces cerevisiae*, Ace1 drives the expression of Cu/Zn SOD [76] and the metallothionine-related genes CSR5 [77] and CUP1 [78] which can bind and buffer excess copper ions. Unfortunately, no metallothionine or metallothionine-related genes are predicted for *P. destructans* within the JGI annotated databank, making it difficult to draw conclusions about this classical Cu-overload biomarker. However, for Cu/Zn SOD (i.e., *VC83_07077*), we do not observe any significant increase in transcript number upon the Cu-overload growth conditions tested in our study. This may indicate that the additional Cu concentration (i.e., 500 μM CuSO_4_) used may not be suitable for eliciting the activation of an ACE1 homolog in *P. destructans*, despite yielding a distinct change in transcripts as indicated in multi-dimensional scaling analysis (Figure 2b). Induction of CUP1/CRS5 in *S. cerevisiae* can be activated in minutes upon Cu stress [14,77]. Thus, our methodology utilizing chronic exposure to Cu may not be suitable for capturing this Cu-adaptation response.

Significant increases in gene transcripts associated with DNA damage and replication stress upon chronic exposure to elevated levels of copper are observed in our Cu-overload datasets. Notably, we observe an increase in transcripts associated with DNA replication, including the genes for the catalytic domain of DNA Polymerase Pol1 and Pol2, *VC83_05453* and *VC83_04573*, respectively. The induction of increased transcripts for *VC83_04943*, a subunit of a replication-pausing checkpoint complex CSM3, suggests that Cu-overloaded cells are experiencing replication stress and assistance in maintaining DNA replication in integrity [79,80]. Additionally, the increased transcripts for *VC83_08538*, which encodes for the ribonucleotide reductase large subunit (RRM1), suggest an active need for dNTP substrates for DNA replication or repair. Ribonucleotide reductase large subunit transcripts have been shown previously to be induced upon exposure to DNA-damaging agents [81]. Our data suggest that *P. destructans* growth under high-copper conditions may significantly impact genomic stability and DNA replication fidelity.

### 4.4. Putative P. destructans Secreted Proteases and Virulence Factors Responsive to Cu Stress

What other putative *P. destructans* virulence factors or genes associated with WNS are responsive to Cu-stress? As *P. destructans* must scavenge all its essential micronutrients from its infected host, our systemic in vitro study provided opportunities to identify genes associated with overcoming Cu-withholding or -overload stress within bat-host infection conditions. We identify a collection of six heat shock proteins (HSPs) or chaperones/co-chaperones that respond to Cu stress and are associated with the WNS state. HSPs in *Candida* [82] and *Histoplasma* [83] species play essential roles in maintaining protein quality control, controlling cellular morphology and drug resistance [84]. Therefore, it is not surprising that infection and nutritional stress lead to elevated transcripts of heat shock and chaperone protein expression.

Early studies on *P. destructans* implemented the secreted collagen-degrading serine protease, Destructin-1 (i.e., *VC83_06062*), as a potential virulence factor to promote fungal tissue invasion [34,85]. Two additional secreted homologs to Destructin-1, namely Destructin-2 (i.e., *VC83_04892*) and Destructin-3 (i.e., *VC83_09074*), were identified as possible proteases of interest [85]. Our results show that both Cu-withholding and Cu-overload stress led to elevated transcripts vs. control for Destructin-1, Destructin-3, and a third serine peptidase *VC83_02181*. This suggests that these secreted factors may respond to general environmental stress. Similarly, elevated transcripts for an additional secreted protease *VC83_02181* also observed upon both Cu-withholding and Cu-overload stress. However, total transcript levels for *VC83_02181* are much less abundant than those for the family of Destructin homologs. This may indicate that *VC83_02181* may play a modest role in the *P. destructans* stress response. Together, our data demonstrate that copper stress can impact the transcription levels of secreted protease and other putative virulence factors associated with the WNS state.

## 5. Conclusions

This report characterizes *P. destructans’* transcriptional response to chronic copper-withholding and copper-overload conditions. Copper-withholding stress leads to the induction of two characteristic pathways that may participate in high-affinity Cu trafficking. The first pathway includes the ubiquitous family of high-affinity Cu transporters and the more recently described Cbi1/Bim1 family of Cu-scavenging proteins. Our data support that both Ctr1a and Ctr1b isoforms, *VC83_00191* and *VC83_048184*, respectively, as well as the two Bim1-like protein (BLP)-encoding genes *VC83_03095* (BLP2) and *VC83_07867* (BLP3), are significantly induced by Cu-withholding stress. *Pseudogymnoascus destructans* may also be capable of utilizing a second high-affinity Cu-trafficking pathway involving secreted metallophore small molecules which are encoded by the genes *VC83_01834*–*VC83_01838*, which we designated the Cu-Responsive gene Cluster (CRC). Future research efforts should be directed at validating the expression levels of CRC genes and identifying metal-binding small molecules in *P. destructans* growth media.

Additionally, our study found that *P. destructans* is resilient to high levels of extracellular copper in its growth media. While we do not observe gross differences in morphology, our data demonstrate that elevated copper leads to elevated transcripts for genes associated with maintaining genome stability and DNA-replication fidelity.

While this study describes the first systemic investigation into the transcriptional response of *P. destructans* to Cu-withholding and Cu-overload stress, it is limited in terms of the functional and metabolic conclusions that can be drawn from transcriptomic analysis alone. As genetic manipulation tools for *P. destructans* become more widely assessable, there will be opportunities to validate and further interrogate the copper cell biology of *P. destructans* to better understand the role of copper in WNS propagation with gene knockout fungal strains.

## Figures and Tables

**Figure 1 jof-11-00372-f001:**
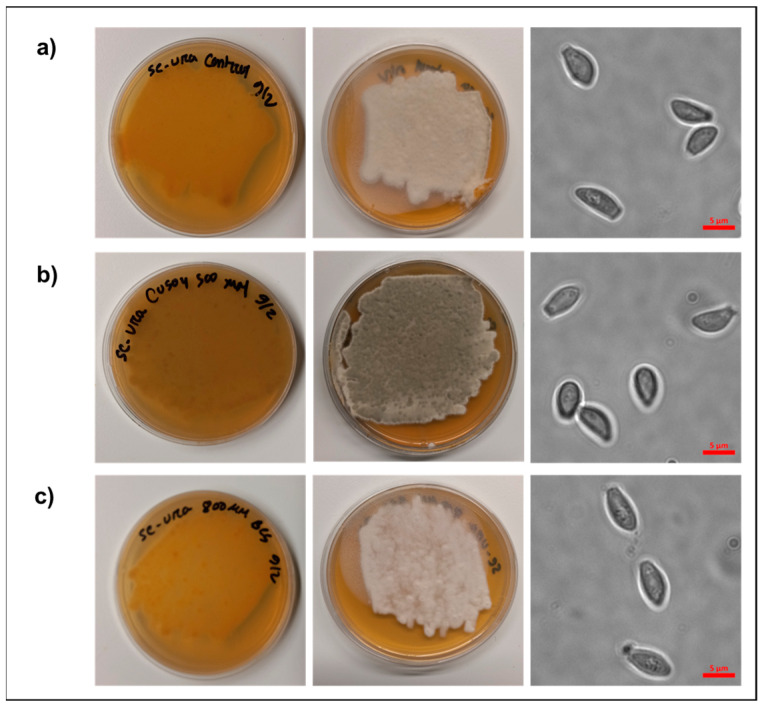
Analysis of *P. destructans* cultures subjected to different Cu-stress growth conditions. *P. destructans* was cultured on (**a**) SC-Ura (control), (**b**) SC-Ura + 500 µM CuSO_4_ (Cu-overload), and (**c**) SC-Ura + 800 µM BCS (Cu-withholding). Right column: Brightfield microscopic images of *P. destructans* spores grown under each condition are displayed at 1000× total magnification.

**Figure 2 jof-11-00372-f002:**
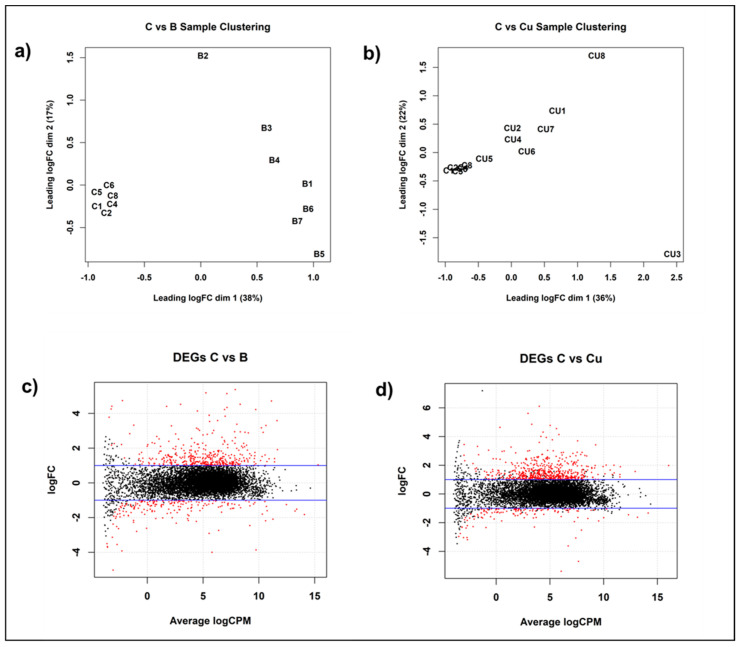
Differential gene expression response of *Pseudogymnoascus destructans* grown under Cu-stress conditions. (**a**) Multi-dimensional scaling (MDS) plot comparing Cu-withholding (“B”, i.e., SC-Ura + 800 µM BCS) treated versus control (“C”, i.e., SC-Ura) samples displaying the relative similarities in gene expression. (**b**) MDS plot comparing Cu-overload (“CU”, i.e., SC-Ura + 500 µM CuSO_4_)-treated versus control (i.e., SC-Ura) samples. (**c**) Genes differentially expressed under Cu-withholding versus control samples. Significantly upregulated and downregulated genes are indicated as red dots. (**d**) Genes differentially expressed under Cu-overload versus control samples. Significantly upregulated and downregulated genes are indicated as red dots.

**Figure 3 jof-11-00372-f003:**
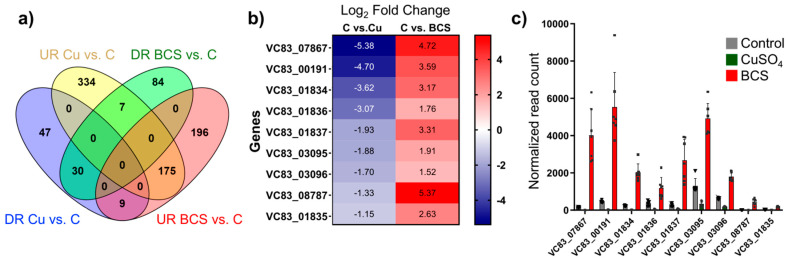
Differential expression of genes in *Pseudogymnoascus destructans* under copper stress conditions. (**a**) Venn diagram showing the differential expression of genes that are significantly upregulated (UR) and downregulated (DR) in *P. destructans* under conditions of Cu-withholding (“BCS”) and Cu-overload (“Cu”) stress. The Venn diagram illustrates the number of genes that are uniquely upregulated or downregulated in each condition, as well as those that are commonly regulated in both stress conditions (*p* < 0.05; Log_2_FC > 1; log_10_CPM > 1). (**b**) List of the (9) DEG shared between DR Cu vs. C and UR BCS vs. C with their corresponding fold changes. (**c**) Normalized read counts of the (9) DEGs shared between DR Cu vs. C and UR BCS vs. C. Plots represent the average read counts and standard error for control (gray bars with triangle symbols; *n* = 6), CuSO_4_ treatment (green bars with circle symbols; *n* = 8), and BCS treatment (red bars with square symbols; *n* = 7).

**Figure 4 jof-11-00372-f004:**
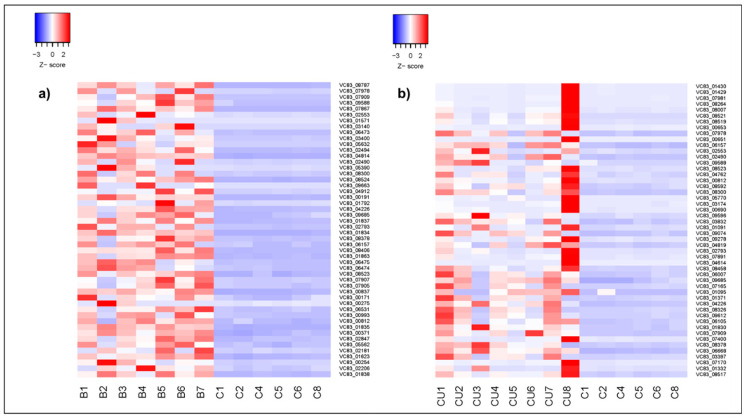
Comparative heat maps of the most abundant 50 differentially expressed genes versus control upon Cu-withholding (**a**) and Cu-overload (**b**) stress conditions. Samples are indicated as C (control, i.e., SC-Ura), B (Cu-withholding;, i.e., SC-Ura + 800 µM BCS), and CU (Cu-overload, i.e., SC-Ura + 500 µM CuSO_4_) *P. destructans* genes using scaled logCPM normalized values. Genes were identified as differentially expressed with a false discovery rate (FDR) of less than 0.001, as determined by edgeR. The log_2_ fold changes (log_2_FC) for these genes ranged from (**a**) 2 to 5 under the BCS condition and (**b**) 2 to 6 under the CuSO_4_ condition.

**Figure 5 jof-11-00372-f005:**
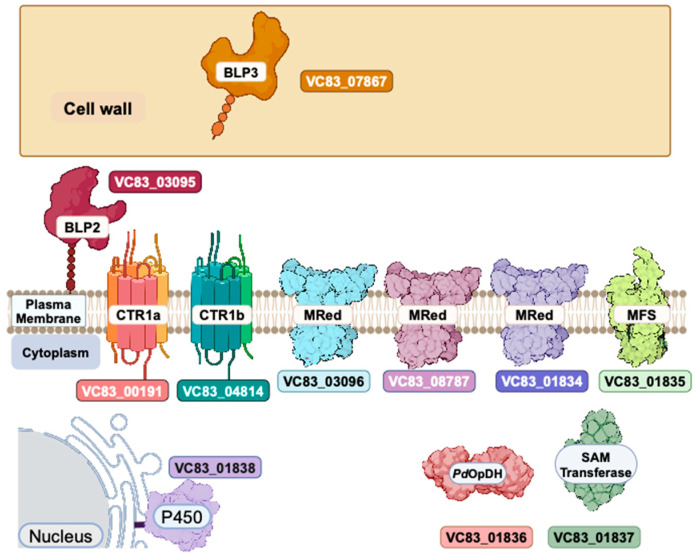
Model displaying the predicted subcellular localization of putative proteins involved in trace metal uptake under Cu-withholding stress conditions. See the text for more details.

**Table 1 jof-11-00372-t001:** Selected gene ontology biological processes of *P. destructans* under Cu-withholding stress conditions.

GotermID	goACC	Biological Process	GOsize ^1^	Fold Enrichment ^2^	*p*-Value ^3^	DEG ^4^
4666	GO:0006536	Glutamate metabolic process	1	16.50	0	1
4712	GO:0006582	Melanin metabolic process	1	16.50	0	1
9015	GO:0015703	Chromate transport	1	16.50	0	1
17658	GO:0042128	Nitrate assimilation	1	16.50	0	1
9010	GO:0015698	Inorganic anion transport	1	16.50	0	1
6710	GO:0009102	Biotin biosynthetic process	1	16.50	0	1
19608	GO:0045132	Meiotic chromosome segregation	1	16.50	0	1
4918	GO:0006801	Superoxide metabolic process	4	12.37	1.34 × 10^−5^	3
4283	GO:0006118	Electron transport	104	2.37	0.0004	15
6276	GO:0008643	Carbohydrate transport	10	4.95	0.0020	3
4851	GO:0006725	Aromatic compound metabolic process	10	4.95	0.0020	3
6724	GO:0009116	Nucleoside metabolic process	2	8.25	0.0036	1
4970	GO:0006857	Oligopeptide transport	2	8.25	0.0036	1
24525	GO:0051276	Chromosome organization and Biogenesis	2	8.25	0.0036	1
5077	GO:0006979	Response to oxidative stress	6	5.50	0.0038	2
4977	GO:0006865	Amino acid transport	8	4.12	0.0098	2
4940	GO:0006825	Copper ion transport	3	5.50	0.0105	1
24507	GO:0051258	Protein polymerization	3	5.50	0.0105	1
5112	GO:0007017	Microtubule-based process	3	5.50	0.0105	1
9784	GO:0016575	Histone deacetylation	3	5.50	0.0105	1
6791	GO:0009186	Deoxyribonucleoside diphosphate Metabolic process	3	5.50	0.0105	1
9582	GO:0016311	Dephosphorylation	9	3.66	0.0141	2
5113	GO:0007018	Microtubule-based movement	10	3.30	0.0192	2
24310	GO:0051056	Regulation of small GTPase mediated signal transduction	4	4.12	0.0202	1

^1^ The number of genes in the GO category represented in the background size; ^2^ fold enrichment, indicating the degree of over-representation of differentially expressed genes; ^3^
*p*-Value indicating the statistical significance of the over-representation of genes in the GO; ^4^ the number of differentially expressed genes (DEGs) in this category identified by *p*-value < 0.05.

**Table 2 jof-11-00372-t002:** Selected gene ontology biological processes of *P. destructans* Cu-overload stress conditions.

GotermID	goACC	Biological Process	GOsize ^1^	Fold Enrichment ^2^	*p*-Value ^3^	DEG ^4^
5073	GO:0006974	Response to DNA damage stimulus	1	14.42	0	1
22806	GO:0048478	Replication fork protection	1	14.42	0	1
4601	GO:0006467	Protein thiol-disulfide exchange	1	14.42	0	1
5240	GO:0007155	Cell adhesion	1	14.42	0	1
4712	GO:0006582	Melanin metabolic process	1	14.42	0	1
267	GO:0000272	Polysaccharide catabolic process	1	14.42	0	1
4599	GO:0006465	Signal peptide processing	2	14.42	0	2
92	GO:0000087	M phase of mitotic cell cycle	1	14.42	0	1
5135	GO:0007040	Lysosome organization and biogenesis	1	14.42	0	1
4792	GO:0006665	Sphingolipid metabolic process	1	14.42	0	1
5022	GO:0006916	Anti-apoptosis	1	14.42	0	1
4199	GO:0006031	Chitin biosynthetic process	1	14.42	0	1
4689	GO:0006559	L-phenylalanine catabolic process	1	14.42	0	1
4700	GO:0006570	Tyrosine metabolic process	1	14.42	0	1
17658	GO:0042128	Nitrate assimilation	1	14.42	0	1
9010	GO:0015698	Inorganic anion transport	1	14.42	0	1
4392	GO:0006231	dTMP biosynthetic process	1	14.42	0	1
24016	GO:0050757	Thymidylate synthase biosynthetic process	1	14.42	0	1
6681	GO:0009072	Aromatic amino acid family metabolic process	1	14.42	0	1
19608	GO:0045132	Meiotic chromosome segregation	1	14.42	0	1
4429	GO:0006270	DNA replication initiation	6	7.21	0.0003	3
4283	GO:0006118	Electron transport	104	2.21	0.0007	16
6276	GO:0008643	Carbohydrate transport	10	4.32	0.0034	3
4421	GO:0006260	DNA replication	16	3.60	0.0036	4
5143	GO:0007049	Cell cycle	2	7.21	0.0048	1
4144	GO:0005976	Polysaccharide metabolic process	2	7.21	0.0048	1
4970	GO:0006857	Oligopeptide transport	2	7.21	0.0048	1
4928	GO:0006811	Ion transport	2	7.21	0.0048	1
24525	GO:0051276	Chromosome organization and biogenesis	2	7.21	0.0048	1
4484	GO:0006334	Nucleosome assembly	13	3.32	0.0098	3
4940	GO:0006825	Copper ion transport	3	4.81	0.0137	1
5112	GO:0007017	Microtubule-based process	3	4.81	0.0137	1
24507	GO:0051258	Protein polymerization	3	4.81	0.0137	1
9350	GO:0016051	Carbohydrate biosynthetic process	3	4.81	0.0137	1
4977	GO:0006865	Amino acid transport	8	3.61	0.0142	2

^1^ The number of genes in the GO category represented in the background size; ^2^ fold enrichment, indicating the degree of over-representation of differentially expressed genes; ^3^
*p*-Value indicating the statistical significance of the over-representation of genes in the GO; ^4^ The number of differentially expressed genes (DEG) in this category identified by *p*-value < 0.05.

**Table 3 jof-11-00372-t003:** Selected protease genes and putative virulence factors that are differential expressed in *P. destructans* under Cu-stress conditions (see text for more details).

Control vs. BCS					
Gene ID	Conserved Domains	Log_2_FC	logCPM	*p*-Value	FDR
VC83_02553	Alpha-crystallin domain (HSP)	4.52	8.32	1.16 × 10^−8^	4.36 × 10^−7^
VC83_00970	Heat shock protein 78, mitochondrial	1.14	7.74	3.75 × 10^−5^	0.0004443
VC83_01046	Heat shock protein 78, mitochondrial	1.21	11.40	4.61 × 10^−5^	0.0005284
VC83_08187	Heat shock protein 78, mitochondrial	1.60	10.55	5.41× 10^−6^	8.85 × 10^−5^
VC83_06435	Hsp90	1.39	8.27	1.37× 10^−5^	0.0001892
VC83_01360	Zip; ZIP Zinc transporter	1.48	5.58	3.15 × 10^−6^	5.59 × 10^−5^
VC83_00191	Ctr; Ctr copper transporter family	3.58	11.62	1.54 × 10^−57^	3.70 × 10^−54^
VC83_04094	ATX1;HMA; Heavy-metal-associated domain (HMA)	1.17	7.67	1.22 × 10^−7^	3.53 × 10^−6^
VC83_07867	BLP3	4.71	11.12	2.13 × 10^−68^	1.03 × 10^−64^
VC83_00261	Endo Mannanase, GH76 Family	1.31	6.56	0.0013	0.0087237
VC83_05292	N/A	1.02	9.59	1.38 × 10^−5^	0.0001895
VC83_06039	Ascorbate peroxidases and cytochrome C peroxidases	1.42	5.80	4.94 × 10^−10^	2.73 × 10^−8^
VC83_01624	ABC-type multidrug transport system	1.70	6.78	3.02 × 10^−15^	5.10 × 10^−13^
VC83_09074	Destructin-3	1.46	4.99	0.0003	0.0030929
VC83_02181	Peptidases_S53	2.51	2.79	2.53 × 10^−7^	6.44 × 10^−6^
**Control vs. Cu**					
VC83_02553	Alpha-crystallin domain (HSP)	3.70	7.57	1.59 × 10^−6^	0.0001101
VC83_01046	Hsp70 chaperone	1.35	11.51	0.0002	0.0054754
VC83_08187	Alpha-crystallin domain (HSP)	1.66	10.60	2.65 × 10^−5^	0.0009256
VC83_06435	STI1 Hsp90 cochaperone	1.24	8.16	8.33 × 10^−5^	0.0022095
VC83_00191	Ctr; Ctr copper transporter family	−4.69	7.65	1.20 × 10^−45^	3.84 × 10^−42^
VC83_04094	ATX1; HMA; Heavy-metal-associated domain (HMA)	1.53	7.94	3.65 × 10^−11^	1.30 × 10^−8^
VC83_07867	BLP3	−5.38	6.04	1.65 × 10^−145^	1.58 × 10^−141^
VC83_09076	Pectate_lyase_	1.60	6.45	1.98 × 10^−10^	6.13 × 10^−8^
VC83_06039	Ascorbate peroxidases and cytochrome C peroxidases	1.59	5.94	0.0012	0.0148732
VC83_08771	MFS_1; Major Facilitator Superfamily	−1.05	6.82	4.13 × 10^−8^	5.45 × 10^−6^
VC83_09074	Destructin-3	2.90	6.24	1.35 × 10^−13^	9.26 × 10^−11^
VC83_06062	Destructin-1	2.01	10.95	0.0022	0.0227017
VC83_02181	Peptidases_S53	2.43	2.74	0.0007	0.0111273

## Data Availability

Raw data supporting the conclusions of this article will be made available by the authors on request.

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
