# Peer review of "Pseudogymnoascus destructans Transcriptional Response to Chronic Copper Stress"

_jof, 2025, doi:10.3390/jof11050372_

Round 1

Reviewer 1 Report

I have no major comments 

Line 110 mentions a concentration of copper ion and copper chelator (125 mM), it is understood that this is the concentration, I suggest mentioning the final concentrations as in line 126.

Figure 1 line 217, I consider that in the result section or the Discussion section, it should be mentioned why the cultures are melanizing in the presence of copper (fig1b), reference where you can find information that may be useful to you (Upadhyay S, Torres G, Lin X. 2013). The microscopic image of the mycelium morphology could be included, it is not mentioned if it had any morphological change at this stage.

Author Response

See attached Document

Reviewer 2 Report

Dear Authors,

the manuscript submitted for review has clear publication potential. Both the research topic and the methodology are highly relevant and up-to-date. However, in my opinion, the manuscript requires certain revisions and clarifications. The work is highly descriptive — at times, it resembles a compilation of results without sufficient in-depth interpretation. It would be beneficial to more clearly connect the transcriptomic findings, for example, with the physiology of the pathogen or its behavior in its natural environment.

No functional experiments are included that would confirm the roles of selected genes (e.g., knock-out/knock-down mutants or growth assays under metal-induced stress conditions). This limits the interpretive strength of the findings. However, if the absence of functional data (e.g., at the protein or metabolomic level) was intentional and such experiments cannot be included at this stage, this should be explicitly acknowledged as a limitation. This would help ensure that the proposed hypotheses are not perceived as more conclusive than the data allow.

A major strength of the manuscript is its broad evolutionary and comparative perspective. At the same time, the scientific style is occasionally too informal — for instance, through the use of expressions such as “we hypothesize.” In scientific writing, it is generally preferable to avoid direct use of “we” when presenting hypotheses, as this may sound overly subjective. Using the passive voice or more impersonal constructions tends to maintain the formal tone appropriate for academic publications.

Additionally, some paragraphs, especially in the Discussion section, are overly long and would benefit from more concise phrasing. I encourage the authors to review this section carefully.

My comments are intended as suggestions for deeper reflection on the manuscript, and they do not preclude its publication. I believe that with revision and refinement, the manuscript has strong potential for acceptance.

I included my comments in the main review.
